# Bilateral Transverse Mandibular Second Molars: A Case Report

**DOI:** 10.3390/dj4040043

**Published:** 2016-11-22

**Authors:** James Puryer, Tarun Mittal, Catherine McNamara, Tony Ireland, Jonathan Sandy

**Affiliations:** 1School of Oral and Dental Sciences, Bristol Dental Hospital, Lower Maudlin Street, Bristol BS1 2LY, UK; tarun.mittal@bristol.ac.uk (T.M.); tony.ireland@bristol.ac.uk (T.I.); jonathan.sandy@bristol.ac.uk (J.S.); 2HSE Regional Orthodontic Department, St James’s Hospital, Dublin 8, Ireland; catherinem.mcnamara1@hse.ie

**Keywords:** bilateral, transverse, impacted, mandibular, molars

## Abstract

Impaction of mandibular second permanent molars is a rare occurrence, with prevalence rates reported to be between 0.65% and 2.0%. In the absence of systemic conditions, impactions are usually unilateral. There appears to be no consensus as to the optimal treatment for impacted mandibular second molars and treatment plans will be based upon the individual case. Treatment may involve orthodontics and/or various surgical techniques, and early diagnosis is important. This paper presents an unusual case of bilateral transverse impaction of both mandibular second and third molars that was diagnosed at 18 years of age. All impacted molars were extracted.

## 1. Introduction

Tooth eruption is defined as “the axial or occlusal movement of a tooth from its developmental position within the jaw towards its functional position at the occlusal plane” [1]. Both systemic and local factors can affect this eruptive process. Systemic factors will often affect multiple teeth and may be present in patients with syndromes, such as cleidocranial dysplasia [2]. Local factors influencing the eruptive process may result in only one or few teeth being affected, usually mandibular third molars and maxillary canines. Disturbances to the eruptive process can have various outcomes including ectopic eruption, impaction, primary retention and secondary retention [3]. Impaction is the cessation of tooth eruption caused by a clinically or radiographically detectable physical barrier in the eruption path. This may be related to physical obstruction in the form of dental crowding, supernumerary teeth, odontomes and odontogenic tumours, or due to an ectopic eruption pathway [4].

Eruptive disturbances affecting second molars are rare [5,6], and the prevalence of mandibular second molar impaction has been reported by various studies to be 0.65%–2.0% [7,8,9,10]. These impactions are commonly mesio-angular in nature. They occur more often unilaterally than bilaterally, in the mandible more often than in the maxilla, in males more than females [11], and are usually associated with an arch length deficiency [12]. Should the increase in arch length not coincide with the eruption of the second molar, the resulting environment will favour impaction. Second molar impaction is usually discovered during orthodontic treatment as an incidental finding [13] and is rarely the reason for the original orthodontic referral [14]. Where it does occur, impaction of the second molar can have a significant clinical impact, affecting a patient’s masticatory ability and aesthetics, as well as increasing the risk of caries or resorption of the distal surface of the first molar [11]. Other pathological outcomes have been reported, including formation of a follicular cyst, pericoronitis, tilting of neighbouring teeth and malocclusion [6,15]. We are unaware of any literature regarding the prevalence of transverse molar impaction.

Due to the potential risks associated with impacted second molars, early diagnosis is important so that treatment options for intervention can be considered. A multi-disciplinary approach may be required and may involve orthodontic treatment and/or surgery, and there appears to be no clear standard method of treating impacted second molars [10]. Various treatment options have been suggested depending upon the diagnosis and position of the second molar [12,16,17,18,19,20,21,22,23,24]. In view of this lack of consensus as to the most appropriate treatment strategy, the uncertain prognosis in the treatment of impacted second molars [25], and the fact that second molars are not vital for efficient mastication [26], all cases need to be assessed on an individual basis and all treatment alternatives should be thoroughly discussed with a patient before intervention.

This paper reports an unusual case of bilateral mandibular second and third molar impaction diagnosed five years following active orthodontic treatment.

## 2. Case Report

A Caucasian female, aged 11 years, with upper and lower arch crowding, was referred to the orthodontic department by her general dental practitioner. She was fit and well with no relevant medical history. The patient had a significant Class II division 1 malocclusion, with severe crowding in both arches. A dental orthopantomogram (OPT) taken at the time of presentation is shown in Figure 1.

Planned treatment involved the extraction of both maxillary first premolars (14, 24) and the left mandibular lateral incisor (32) followed by upper and lower fixed appliance therapy. This treatment was carried out to reach a favourable post-treatment occlusion, and the fixed appliances were removed approximately 18 months later when the patient was aged 13 years. The patient was given removable retainers, and placed on a review programme.

A five year review was carried out when the patient was aged 18-years, and clinical examination showed that both lower second molars (37, 47) had not yet erupted. An OPT (Figure 2) was taken which showed that both mandibular second and third molars (37, 38, 47, 48) had developed a transverse impaction with little chance of natural eruption. There was no sign of resorption of the mandibular first molars (36, 46). A further course of orthodontic treatment to upright the second molars was not considered to be a viable option at this stage and the four transversely impacted lower molars (37, 38, 47, 48) were surgically extracted, along with the maxillary third molars (18, 28). The extractions were carried out by a consultant maxillofacial surgeon in hospital under general anaesthetic. No complications arose during the surgery, and healing was uneventful.

## 3. Discussion

This paper presents a very rare case of bilateral transverse impaction of both mandibular second and third molars. It is unusual in that multiple teeth are affected in a patient with no relevant medical history or syndromes. There was no indication of the future impaction of these teeth when the original radiograph was taken aged 11 years, and like many impacted second molars, these were only identified radiographically when it transpired that the teeth had still not erupted well beyond their normal eruption dates.

The reasons for these impactions are unclear. There were no obvious physical barriers to prevent the eruption of these teeth, and it seems likely that all four teeth had an ectopic eruption pathway [4]. This patient did have significant crowding which is a known aetiological factor [27].

We speculate that, had a radiograph been taken at aged 13, it may have been seen that the lower second molars were erupting into an abnormal position, and there may have been opportunity to correct this. However, this practice would go against current radiographic guidelines from both the British Orthodontic Society (BOS) [28] and Faculty of General Dental Practice (UK) [29] that were followed throughout this case. The BOS guidelines state that “The clinical justification for radiographs after treatment or at the end of retention is difficult to define and has to be assessed for each patient” [28]. This guidance is similar to that issued in both the USA and Europe. The American Dental Association recommends that “clinical judgement be used in determining the need for, and type of radiographic images necessary for, evaluation and/or monitoring of dentofacial growth and development” [30]. The European Commission guidelines on Radiation Protection in Dental Radiology state “the need for radiography to monitor treatment progress is dependent upon a careful clinical assessment” [31]. As the previous radiograph taken at age 11 years showed all permanent teeth to be developing correctly, and that second molars are often not erupted at age 13 years, there was no clinical indication to take further radiographs after completion of active treatment. During the retention period from 13 years onwards, it would have become apparent that the mandibular second molars were not erupting. Thus, it would have been appropriate to undertake further radiographic assessment based upon clinical assessment and individual need given that the second molars had not erupted by the age of 14 years. The authors were not involved in the care of this patient during this review period, and so the reason a follow-up radiograph was not taken is unknown. There was no resorption of the distal surfaces of the first molars associated with these impactions which, if present, would have made the outcome of this unpredictable occurrence more significant.

## 4. Conclusions

This paper presents an unusual case of bilateral transverse impaction of both mandibular second and third molars. These impactions were diagnosed at 18 years of age, and because of the degree of impaction, all four teeth were extracted. This case highlights the need for clinicians to follow radiographic guidelines. Despite orthodontic treatment having been carried out successfully, this unusual outcome may possibly have been prevented.

## Figures and Tables

**Figure 1 dentistry-04-00043-f001:**
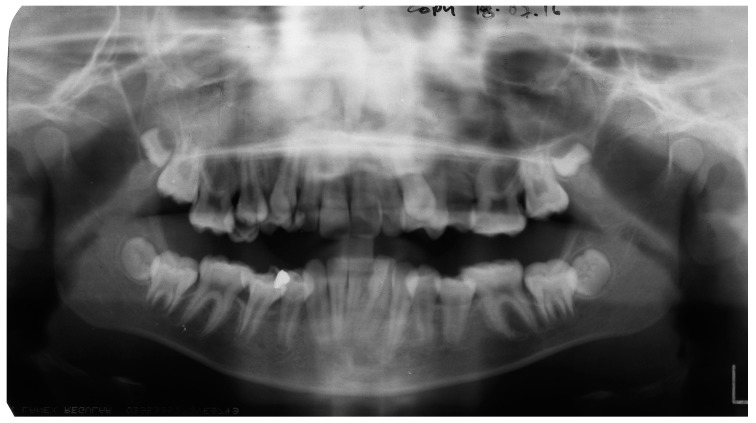
A dental orthopantomogram (OPT) of patient aged 11 years.

**Figure 2 dentistry-04-00043-f002:**
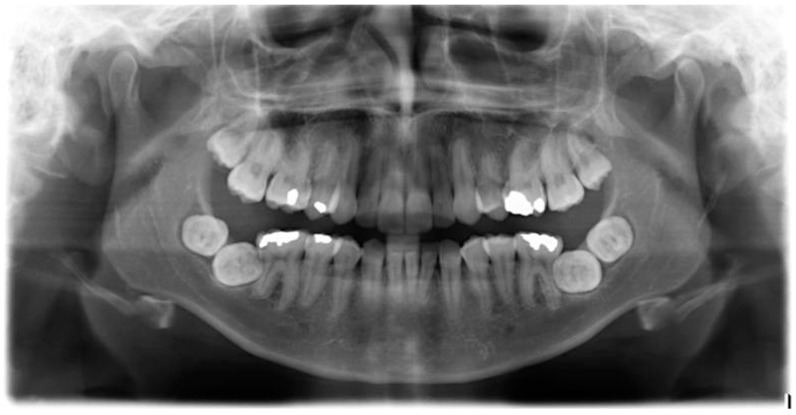
OPT of patient aged 18 years.

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
