# Peer review of "Bilateral Transverse Mandibular Second Molars: A Case Report"

_dentistry, 2016, doi:10.3390/dj4040043_

Round 1
Reviewer 1 Report
The research presents an unusual case of occlusion of second molars and its introduction and references are appropriate.
the overall conduct of the clinical case is questionable due to the possible lack of post-orthodontic treatment controls, although it can be assumed that the authors were not directly involved in conducting the orthodontic treatment. If this is the case, the situation must be explained in the text.
Some details of the surgery and intraoperative observations may be useful for the reader.
Author Response
Thank you for this very positive review of our paper and detailing how it may be improved.
We feel that we have been able to address your comments by:
- explain that the authors were not directly involved in the care of the patient
- discussed the review period between 13 and 18 years in more detail and related this to radiographic guidelines, explaining that a radiograph ideally would have been taken earlier than at age 18 years.
- added a brief description of how surgery was undertaken
We hope that you will look favourably on these amendments.
Reviewer 2 Report
This paper addresses interesting issues regarding the morphological anomalies and congenitally missing in the daily clinical practice. However it requires some amendments in the manuscript. The followings are some amendments and additions need to be considered.
Introduction;
The aim and the rationale for the case report were clearly outlined in the part of the introduction.
Case report, Discussion, and Conclusion;
During the five year review (follow up) of the patient, anomalies in the eruption of lower second and third molars should be aware by authors. Regarding the BOS guidelines, interpretation of the meaning of “has to be assessed for each patient” would be very important. How do authors think the timing of taking radiographs? Following the BOS guidelines, each practitioner has their own guidelines as well. If the authors express their own guidelines in the discussion part, there would be beneficial information for readers. How do authors think about the diagnostic criterion regarding the delayed eruption of second molars? More detailed description would be needed.
As the useful information for readers, explanation of the radiographic guidelines of the other countries would be needed in the discussion part.
References;
The References were up-to-date and relevant.
Author Response
Thank you for your overall positive review of our paper and for suggesting how it may be improved.
We feel that we have been able to address your comments by:
- revising the Discussion part of the paper to relate the review period of the patient from 13 to 18 years to radiographic guidance. We have hopefully clarified that ideally a further radiograph would have been taken at an earlier stage when it was noticed that the second molars had not erupted by 14 years of ago.
- revising the Conclusions to reflect this also.
- adding to the Discussion part of the paper further guidance from the USA and Europe, along with two appropriate references.
We hope that you will approve of these revisions and be supportive of publication of our paper in its revised form.
Round 2
Reviewer 1 Report
The authors responded to the objections presented and the majority of them have been addressed.
Reviewer 2 Report
Since all comments and objections have now been answered satisfactorily, I recommend that this paper is published.